# PVR and ICAM-1 on Blast Crisis CML Stem and Progenitor Cells with TKI Resistance Confer Susceptibility to NK Cells

**DOI:** 10.3390/cancers12071923

**Published:** 2020-07-16

**Authors:** Nayoung Kim, Mi-Yeon Kim, Young-Uk Cho, WenYong Chen, Kyoo-Hyung Lee, Hun Sik Kim

**Affiliations:** 1Department of Convergence Medicine, University of Ulsan College of Medicine, Seoul 05505, Korea; naykim@amc.seoul.kr; 2Asan Institute for Life Sciences, Asan Medical Center, University of Ulsan College of Medicine, Seoul 05505, Korea; 3Department of Biomedical Sciences, University of Ulsan College of Medicine, Seoul 05505, Korea; sunkimnkt@daum.net; 4Department of Laboratory Medicine, University of Ulsan College of Medicine and Asan Medical Center, Seoul 05505, Korea; yucho@amc.seoul.kr; 5Department of Cancer Biology, Beckman Research Institute, City of Hope, Duarte, CA 91010, USA; wechen@coh.org; 6Department of Hematology, University of Ulsan College of Medicine and Asan Medical Center, Seoul 05505, Korea; khlee2@amc.seoul.kr; 7Microbiology, University of Ulsan College of Medicine, Seoul 05505, Korea; 8Stem Cell Immunomodulation Research Center (SCIRC), Asan Medical Center, University of Ulsan College of Medicine, Seoul 05505, Korea

**Keywords:** chronic myeloid leukemia, blast crisis, natural killer cells, leukemic stem cells

## Abstract

The *BCR-ABL1* fusion gene generating an oncogenic tyrosine kinase is a hallmark of chronic myeloid leukemia (CML), which can be successfully targeted by BCR-ABL1 tyrosine kinase inhibitors (TKIs). However, treatment-free remission has been achieved in a minority of patients due to evolving TKI resistance and intolerance. Primary or acquired resistance to the approved TKIs and progression to blast crisis (BC), thus, remain a major clinical challenge that requires alternative therapeutic strategies. Here, we first demonstrate that donor natural killer (NK) cells prepared using a protocol adopted in clinical trials can efficiently eliminate CML-BC blasts, with TKI resistance regardless of *BCR-ABL1* mutations, and preferentially target CD34^+^CD38^−^ leukemic stem cells (LSC), a potential source of disease relapse. Mechanistically, the predominant expression of PVR, a ligand for the NK cell-activating DNAM-1 receptor, in concert with ICAM-1, a ligand for NK cell adhesion, confer this susceptibility to NK cells, despite the lack of ligands for NKG2D, a principal NK cell activating receptor, as an immune evasion mechanism. With these mechanistic insights, our findings provide a proof-of-concept that donor NK cell-based therapy is a viable strategy for overcoming TKI resistance in CML, particularly the advanced, multi-TKI-resistant CML with dismal outcome.

## 1. Introduction

Chronic myeloid leukemia (CML) results from the malignant transformation of a hematopoietic stem cell, caused by oncogenic BCR-ABL1 fusion proteins with constitutive tyrosine kinase activity. CML usually presents at the chronic phase (CP) but, if left untreated or standard therapy fails, progresses to an accelerated phase and a terminal blast crisis (BC) with poor prognosis. The current treatment for CML is tyrosine kinase inhibitor (TKI) targeting BCR-ABL1, including imatinib (IM), nilotinib, and dasatinib [1]. Despite remarkable clinical responses with TKIs, the primary or acquired TKI resistance and progression to BC are major challenges limiting treatment options. TKI resistance usually occurs through the acquisition of point mutations in the BCR-ABL1 kinase domain [2]. Among the BCR-ABL1 mutations identified in relapsed CML patients, T315I mutation is frequent (20–30%) and unresponsive to all approved BCR-ABL1 TKIs, except ponatinib, being of limited use due to dose-dependent cardiovascular toxicity. However, compound mutations (≥2 mutations in cis) confer resistance even to ponatinib. TKI resistance also results from BCR-ABL1-independent mechanisms, involving the activation of alternative oncogenic pathways without a detectable BCR-ABL1 mutation or disease-initiating leukemic stem cells (LSC) that are resistant and retained even in most TKI responders [3,4]. Thus, more rational therapeutic strategies would be required to circumvent problems related to TKI resistance and intolerance.

Natural killer (NK) cells are major effectors in cancer immunosurveillance and considered promising cancer therapeutics due to their inherent cancer selectivity [5,6]. Using an array of receptors responding to cellular transformation, NK cells are hard-wired to recognize and eliminate a broad spectrum of cancer cells, including cancer stem cells [7], via direct cytolysis and IFN-γ production. NK cell effector function is triggered by diverse activating receptors that recognize specific ligands on target cells, including NKG2D, NKp30, NKp44, NKp46, and DNAM-1. This activation is kept in check by inhibitory receptors, such as KIRs and CD94-NKG2A, specific for MHC class I molecules frequently downregulated on cancer cells. Moreover, the binding of LFA-1 to ICAM on target cells is a common requirement for NK cell adhesion to target cells for efficient cytolysis [8]. In CML, the frequency and proliferative capacity of NK cells are significantly reduced upon disease progression [9], whereas increased mature NK cells characterize patients with relapse-free survival following imatinib discontinuation [10]. Thus, given the variety and versatility of cancer-targeting NK receptors, NK cell-based therapy may benefit CML patients, particularly with TKI resistance. Thus, this study was conducted to investigate the feasibility of NK cells in targeting CML-BC with a focus on BCR-ABL1 mutations.

## 2. Results and Discussion

### 2.1. CML-BC Cells with BCR-ABL1 T315I Mutation are Susceptible to NK Cell-Mediated Killing

To investigate the effect of NK cells on CML-BC cells sensitive or resistant to IM, we used KCL-22M cells and human CML-BC cells harboring a T315I mutation versus parental KCL-22 cells [11]. The T315I mutation was confirmed by direct sequencing of genomic DNA from KCL-22M cells (Appendix A) [11]. As expected, KCL-22M cells, but not KCL-22 cells, were resistant to apoptosis following IM treatment (Figure 1a). To assess their susceptibility to NK cells, primary NK cells from healthy donors expanded using cytokines and feeder cells, a protocol adopted in clinical trials for hematological malignancies, were used as effector cells. NK cells efficiently killed KCL-22M cells, which were unaffected by IM pretreatment (Figure 1b). By comparison, IM-pretreated KCL-22 cells undergoing apoptosis were less susceptible to NK cells. Consistently, NK cell degranulation and IFN-γ production were well-maintained against KCL-22M cells, regardless of IM pretreatment, but significantly reduced against IM-pretreated KCL-22 cells (Figure 1c,d) and, similarly, IM-sensitive K562 cells (Appendix A). Thus, these results suggest a therapeutic potential of donor NK cells in the elimination of CML-BC cells resistant, but not sensitive, to TKI.

### 2.2. NK Cell Cytotoxicity Correlates with Activating Ligand Expression after IM Treatment

We next analyzed surface expression of ligands for NK activating receptors to understand mechanisms underlying the dichotomic pattern of susceptibility to NK cells. IM treatment led to dose-dependent downregulation of NKG2D ligands (NKG2DL), such as MICA/B, ULBP1, ULBP2/5/6, and ULBP3 on KCL-22 cells (Figure 2a), consistent with IM-mediated decrease in NKG2DL on K562 cells and their reduced lysis by NK cells [12]. We also found similar downregulation of DNAM-1L (CD155/PVR and CD112/Nectin-2) and NKp30L (B7-H6). However, their expression was well-preserved on KCL-22M cells after IM treatment (Figure 2b), likely accounting for the retained susceptibility of KCL-22M cells to NK cells. This notion was supported by the significant contribution of NKG2D, DNAM-1, and, to a lesser extent, NKp30 and 2B4 to NK cytotoxic degranulation, as assessed by the receptor blockade (Figure 2c).

### 2.3. NK Cells Exert Cytotoxicity against Primary CD34^+^ CML-BC Cells with Different BCR-ABL Mutation

We further analyzed primary CML-BC cells to understand the clinical implications of the findings from KCL-22M cells. Among seven TKI-resistant CML-BC patients, four patients harbored a single mutation (T315I, E255V, or E255K), one had compound mutation (F317L/E275K), and two had no mutation in the *BCR-ABL1* gene (Appendix A). NK cells consistently triggered apoptosis of CD34^+^ CML-BC cells in a time-dependent manner, regardless of BCR-ABL1 mutation, as assessed by annexin-V staining (Figure 3a,b). In support, NK cell degranulation correlated with apoptosis of CML-BC cells (Appendix A). We next assessed the expression of activating ligands on CML-BC cells, given its association with NK cytotoxicity. Unlike the KCL-22M cell line, NKG2DL and NKp30L expression were absent or weak, whereas CD155/PVR among DNAM-1L was prominently expressed (Figure 3c,d). Accordingly, apoptosis of CML-BC cells was significantly impaired by the blockade of DNAM-1, but not NKG2D and NKp30 (Figure 3e,f). These results suggest NKG2DL downregulation at CML-BC compared with CML-CP [12,13] as an immune evasion mechanism, and highlight the potential of DNAM-1-directed therapy for advanced phase CML. In this regard, a comparison study between TKI-sensitive and -resistant CML blasts with respect to their ligand expression and susceptibility to NK cells merits further investigation.

### 2.4. Association of ICAM-1 with Increased Susceptibility of CML-BC LSC to NK Cells

CD34^+^ CML blasts include primitive CD34^+^CD38^−^ cells containing LSC and CD34^+^CD38^+^ progenitor cells. Importantly, CD34^+^CD38^−^ LSC are resistant to TKIs, regardless of BCR-ABL1 activation, accounting for disease persistence and relapse [14]. Accordingly, we assessed NK cytotoxicity against CD34^+^CD38^+^ versus CD34^+^CD38^−^ subsets. CML-BC blasts from four patients were identified to contain both subsets (Appendix A), compatible with the reduced expression of CD38 in CML-BC CD34^+^ blasts (i.e., increased frequency of CD34^+^CD38^−^ LSC) [15], and thus, used for further experiments. Notably, CD34^+^CD38^−^ LSC were more susceptible than CD34^+^CD38^+^ progenitor cells to NK cells, although their lyses by NK cells were both significant (Figure 4a,b). To probe mechanisms of increased susceptibility, we analyzed the expression of selected ligands for NK cell receptors that control cytotoxicity. CD155/PVR expression was comparable between both subsets (Figure 4c,d). Moreover, HLA-ABC and HLA-E expression for NK inhibitory receptors were marginal and comparable (Figure 4e). Interestingly, ICAM-1, rather than ICAM-2, expression was upregulated on the CD34^+^CD38^−^ subset compared with the CD34^+^CD38^+^ subset (Figure 4e,f), correlating with their susceptibility to NK cells. Accordingly, ICAM-1 blockade significantly reduced the apoptosis of both subsets (Figure 4g,h). ICAM-1 is strongly expressed by “stem cell-derived” malignancies, particularly on CML-BC blasts [16] and the CD34^+^CD38^−^ lymphoblast in childhood ALL [17]. Thus, ICAM-1 upregulation may be an intrinsic feature of CML LSC, given the association between downregulation of adhesion molecules including ICAM-1 and egress of CD34^+^CD38^−^ cells from bone marrow to peripheral blood [18,19]. Although further study is required to fully understand the increased susceptibility, differential expression of ICAM-1 appears critical for regulating NK cytotoxicity against CML-BC blasts.

## 3. Materials and Methods

### 3.1. Cells and Reagents

Primary samples were obtained from CML-BC patients and healthy donors after informed consent in accordance with protocols approved by the Institutional Review Board (IRB) of Asan Medical Center and the Declaration of Helsinki (2013-0850, 16 Sep 2013 and 2018-0611, 28 May 2018). All of the study patients were resistant to TKIs including ponatinib (patient characteristics in Appendix A). Among the seven patients enrolled in this study, five harbored BCR-ABL1 kinase domain mutations, including one case with F317L/E275K compound mutation, whereas the others did not. Mononuclear cells (MNCs) were separated from peripheral blood or BM samples by Ficoll density gradient centrifugation (LSM lymphocyte separation medium; MP Biomedicals). Expanded primary NK cells were used as effector cells after 24 h incubation with human recombinant IL-2 (rIL-2) (200 U/mL; Roche, Basel, Switzerland). K562 cells (ATCC) were cultured in Iscove’s modified Dulbecco’s medium (IMDM; Gibco) supplemented with 10% FBS and 2 mM L-glutamine (Gibco). The human CML-BC cell line KCL-22 was cultured in RPMI1640 (Gibco) supplemented with 10% FBS (Gibco). The human TKI-resistant CML-BC cell line KCL-22M, harboring a BCR-ABL1 T315I, was cultured in RPMI1640 supplemented with 10% FBS, and 2 μM imatinib (Sigma-Aldrich, St. Louis, MO, USA). The cells were confirmed to be free of mycoplasma contamination. The T315 mutation in one allele of the *BCR-ABL1* gene compared with one wild-type allele of the *c-ABL* gene in the KCL-22M cell line was confirmed using polymerase chain reaction (PCR) amplification and direct sequencing of genomic DNA, as previously described [11].

### 3.2. NK Cell Expansion

Primary human NK cells were expanded as previously described [20], with some modifications. PBMCs (1.5 × 10^6^ cells) were incubated in a 24-well tissue culture plate with 100 Gy-irradiated K562-mb15-41BBL cells (1 × 10^6^ cells) in Stem Cell Growth medium (SCGM; CellGenix, Freiburg, German) supplemented with 10% FBS and 10 U/mL rIL-2. The medium was exchanged every 2 days with fresh medium containing 10 U/mL rIL-2. After a week, residual T cells were depleted with a CD3 positive selection kit (StemCell Technologies, Cambridge, MA, USA). The remaining cells were incubated in SCGM supplemented with 10% FBS, 100 U/mL rIL-2, and 5 ng/mL rIL-15 for two additional weeks, with a medium exchange being made every 2 days. The expanded cell populations were 96 to 99% CD3^−^CD56^+^, as assessed using flow cytometry.

### 3.3. Antibodies

Antibodies (Abs) for NK cell receptors and ligands were obtained from the following sources: anti-human NKG2D (1D11), anti-human DNAM-1 (DX11), and anti-human CD335/NKp46 (9E2/NKp46) from BD Biosciences (San Jose, CA, USA); anti-human CD337/NKp30 (210845) from R&D System; anti-human CD244/2B4 (C1.7), anti-human CD337/NKp30 (P30-15), CD336/NKp44 (P44-8), anti-human ICAM-1 (HA58), anti-human ICAM-2 (CBR-IC2/2), and isotype control mouse IgG1 (MOPC-21) from BioLegend; anti-human HLA-E (MEM-E/06) from Santa Cruz. The following fluorochrome-conjugated Abs were used in flow cytometric analyses: anti-human CD107a-FITC (H4A3), anti-human CD3-PerCP (SK7), anti-human CD34-FITC or PE (8G12), anti-human CD56-PE (NCAM16.2), anti-human MICA/B-PE (6D4), anti-human Nectin-2/CD112-PE (R2.525), anti-human ICAM-1-PE (HA58), anti-human ICAM-2-PE (CBR-IC2/2), anti-human HLA-ABC-PE (G46-2.6), isotype control mouse IgG1 conjugated with FITC or PE (MOPC-21), and isotype control mouse IgG2a conjugated with PE (G155-178) from BD Biosciences; anti-human PVR/CD155-PE (SKII.4), anti-human CD38-APC (HB-7), isotype control mouse IgG1 conjugated with APC (MOPC-21), and isotype control mouse IgG2b conjugated with PE (MPC-11) from BioLegend; anti-human ULBP1-PE (170818), anti-human ULBP2/5/6-PE (165903), anti-human ULBP3-PE (166510), and anti-human B7-H6-PE (875001) from R&D System. AffiniPure F(ab’)2 fragment goat anti-mouse IgG-PE was from Jackson ImmunoResearch. Fc Receptor Binding Inhibitor antibody was obtained from eBioscience (San Diego, CA, USA).

### 3.4. NK Cell Degranulation Assay

The cytotoxic degranulation of NK cells was determined by measuring the cell surface expression of CD107a, as previously described [20]. Briefly, primary expanded NK cells (1 × 10^5^ cells) were mixed with an equal number of KCL-22, KCL-22M, K562 cells, or primary TKI-resistant CML-BC blasts in a 96-well V-bottom culture plate (Costar, Corning, NY, USA) and incubated for 2 h at 37 °C. For the blockade of NK activating receptors, primary expanded NK cells were Fc blocked with human Fc Receptor Binding Inhibitor (eBioscience), and then, incubated with 20 μg/mL control IgG1 or Abs to the indicated NK activating receptors for 30 min at 4 °C, prior to mixing with target cells. The cell pellets were resuspended in flow cytometry buffer (phosphate-buffered saline (PBS) with 1% FBS) and stained with anti-human CD3-PerCP, anti-human CD56-PE, and anti-human CD107a-FITC Abs for 30 min in the dark at 4 °C. Lymphocytes were gated on FCS and SSC characteristics, and the CD107a expression on CD3^−^CD56^+^ NK cells was analyzed using flow cytometry (FACS Accuri C6, BD Biosciences) and Tree Star FlowJo software (ver.10, Treestar, Ashland, OR, USA).

### 3.5. NK Cell IFN-γ Production Assay

Primary expanded NK cells (1 × 10^5^ cells) were stimulated with an equal number of KCL-22 or KCL-22M cells for 1 hour at 37 °C. Then, brefeldin A (GolgiPlug; BD Biosciences) was added, and followed by incubation for an additional 5 h, for a total of 6 h. The cells were first stained with anti-human CD3-PerCP and anti-human CD56-PE antibodies for 30 min in the dark at 4 °C. Samples were then washed twice with FACS buffer and incubated in BD Cytofix/Cytoperm solution (BD Biosciences) for 20 min in the dark at 4 °C. The cells were then washed twice with BD Perm/Wash buffer (BD Biosciences), stained with anti-human IFN-γ-FITC for 30 min in the dark at 4 °C, washed again, and analyzed by flow cytometry gated on CD3^−^CD56^+^ NK cells.

### 3.6. Assay of Target Cell Lysis by NK Cells

For the europium-based cytotoxicity assay, KCL-22 or KCL-22M cells pretreated without or with IM (5 μM) were loaded with 40 µM bis (acetoxymethyl) 2,2’:6’,2’’- terpyridine- 6,6’’- dicarboxylate (BATDA) (Perkin Elmer, Waltham, MA, USA) for 30 min at 37 °C. Cells were then washed in a medium containing 1 mM sulfinpyrazone (Sigma, St. Louis, MO, USA) and incubated with primary expanded NK cells in the presence of sulfinpyrazone for 2 h at 37 °C. Plates were mixed briefly and centrifuged at 1400 rpm for 5 min. The supernatant (20 µL) was incubated with 200 µL of 20% europium solution (Perkin Elmer) in 0.3 M acetic acid for 5 min, and target cell lysis was detected using VICTOR X4 multi-label plate leader (Perkin Elmer).

For the annexin V-based apoptosis assay, primary expanded NK cells (1 × 10^5^ cells) were mixed with an equal number of primary TKI-resistant CML-BC blasts with different BCR-ABL mutations in a 96-well V-bottom culture plate (Costar) and incubated for the indicated times at 37 °C. For the blockade of NK activating receptors, primary expanded NK cells were Fc blocked with human Fc Receptor Binding Inhibitor (eBioscience), and then, incubated without or with 20 μg/mL control IgG1 or Abs to the indicated NK activating receptors for 30 min at 4 °C, prior to mixing with target cells. For the blockade of adhesion ligands, primary TKI-resistant CML-BC cells (1 × 10^5^ cells) were Fc blocked as above, and then, incubated without or with 20 μg/mL control IgG1 or Abs to ICAM-1 or ICAM-2 for 30 min at 4 °C, prior to mixing with effector cells. After incubation, the cell pellets were resuspended in flow cytometry buffer and then, stained with anti-human CD34-PE and anti-human CD38-APC Abs for 30 min in the dark at 4 °C. The cells were washed, resuspended in Annexin-V binding buffer (BD Biosciences), and then, stained with annexin V-FITC for 15 min in the dark at room temperature. The annexin-V-positive apoptotic cells were analyzed by flow cytometry gated on CD34^+^, CD34^+^CD38^+^, or CD34^+^CD38^−^ CML-BC blasts.

### 3.7. Statistical Analysis

All the experiments were independently repeated at least two times. Individual data points between two groups were analyzed by two-tailed or one-tailed Student’s *t*-test using the GraphPad Prism software (ver.5.00, GraphPad Software, Inc., San Diego, CA, USA). Statistical significance was defined as *p* < 0.05, and the degree of significance is indicated as follows: * *p* < 0.05, ** *p* < 0.01, and *** *p* < 0.001.

## 4. Conclusions

Altogether, we found that activated donor NK cells efficiently eliminate CML-BC cells, with TKI resistance in the absence or presence of BCR-ABL1 mutations, and preferentially target the LSC-containing CD34^+^CD38^−^ subset, a potential source of disease relapse. Moreover, the finding of PVR and ICAM-1 involvement in such a context provides new insight into the molecular mechanisms of NK cytotoxicity against CML-BC blasts. Given the capacity of NK cells mediating ADCC via CD16, combination therapy with antibodies directed against LSC can be pursued to improve the eradication of CD34^+^CD38^−^ LSC [21,22]. Although further validation is required in a larger cohort of patients and in vivo models, the present study provides proof-of-concept that NK cell-based therapy holds promise as a viable strategy targeting TKI-resistant CML-BC with dismal outcome.

## Figures and Tables

**Figure 1 cancers-12-01923-f001:**
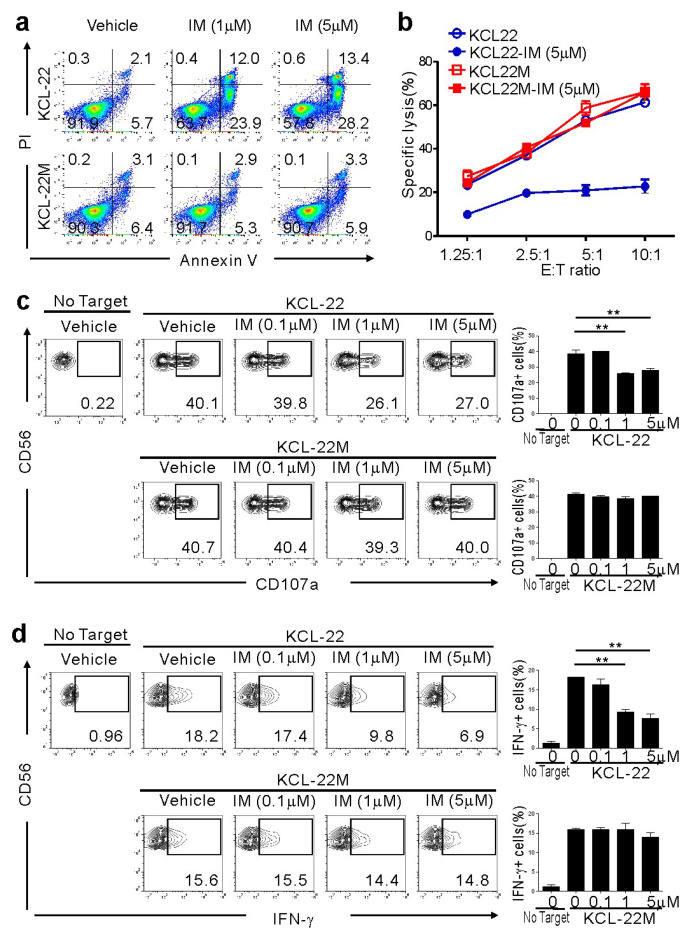
Retained susceptibility of KCL-22M cells, a chronic myeloid leukemia (CML)-blast crisis (BC) cell line harboring BCR-ABL1 T315I, to natural killer (NK) cells following imatinib mesylate (IM) treatment. (**a**) KCL-22 harboring wild-type BCR-ABL1 and KCL-22M cells harboring BCR-ABL1 T315I were treated with IM at the indicated concentrations or vehicle only for 48 h, and apoptosis was measured by annexin V-FITC and PI staining. (**b**) KCL-22 or KCL-22M cells were pretreated with IM (5 μM) or vehicle only for 48 h, washed, and then, incubated with expanded primary NK cells at the indicated effector to target (E:T) cell ratio. Cytotoxicity against the CML-BC cell line was measured after 2 h with the europium assay. (**c**) KCL-22 or KCL-22M cells were pretreated with IM at the indicated concentrations for 48 h, washed, and then, incubated with expanded primary NK cells for 2 h. Degranulation of NK cells was measured by cell surface expression of CD107a on CD3^−^CD56^+^ NK cells. Representative flow cytometry profiles (left) and graph of statistical bar charts (**right**) showing the percent expression of CD107a on NK cells. (**d**) KCL-22 or KCL-22M cells pretreated with IM as in (**c**) were mixed with expanded primary NK cells for intracellular cytokine assay. After 6 h incubation, cells were stained with fluorochrome-conjugated anti-CD3 and anti-CD56 mAbs for surface staining. IFN-γ production was measured in CD3^−^CD56^+^ NK cells after intracellular staining of interferon (IFN)-γ. Representative flow cytometry profiles (**left**) and graph of statistical bar charts (b) showing the percent expression of IFN-γ by NK cells. The mean values ± s.d. of three independent experiments are shown. ** *p* < 0.01 by Student’s *t*-test.

**Figure 2 cancers-12-01923-f002:**
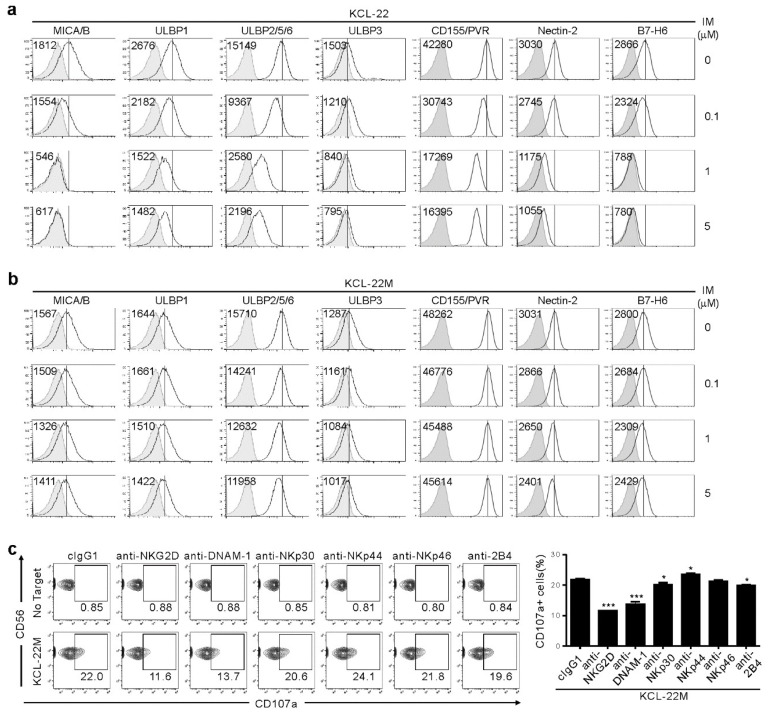
Association of NKG2DL and DNAM-1L expression with the susceptibility to NK cells after IM treatment. (**a**,**b**) KCL-22 (**a**) or KCL-22M cells (**b**) were cultured for 48 h with the indicated doses of IM. Thereafter, cells were analyzed for the surface expression of MICA/B, ULBP1, ULBP2/5/6, and ULBP3 for NKG2D receptor, PVR and Nectin-2 for DNAM-1 receptor, or B7-H6 for NKp30 receptor by flow cytometry. The representative mean fluorescence intensity (MFI) of each ligand (upper left) in the histogram is presented as the result of three independent experiments. The solid lines indicate isotype control staining. (**c**) KCL-22M cells were pretreated with IM (5 μM) for 48 h, washed, and then, mixed with expanded primary NK cells preincubated with mAb to the indicated receptors (20 μg/mL) for 2 h. Degranulation of NK cells was measured by percent expression of CD107a on CD3^−^CD56^+^ NK cells. Representative flow cytometry profiles (**left**) and a graph of statistical bar charts (**right**), demonstrating the percentage of CD107a^+^ NK cells. The mean values ± s.d. of three independent experiments are shown. * *p* < 0.05; *** *p* < 0.001 by Student’s *t*-test.

**Figure 3 cancers-12-01923-f003:**
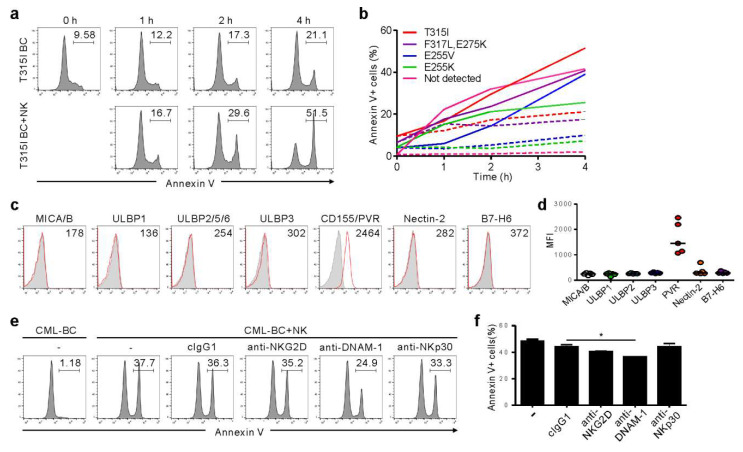
NK cells trigger cytolysis of primary CML-BC blasts, regardless of BCR-ABL1 mutation, in a DNAM-1-dependent manner. (**a**) Mononuclear cells (MNCs) from a CML-BC patient harboring BCR-ABL1 T315I were incubated with expanded primary NK cells for the indicated times. Apoptosis of CD34^+^ CML-BC cells in the absence or presence of NK cells in a 1:1 ratio was measured by annexin V-FITC and CD34-PE staining. The representative percent apoptosis of CD34^+^ CML-BC cells is presented as the result of two independent experiments. (**b**) MNCs from a CML-BC patient harboring different BCR-ABL1 mutations (*n* = 5) were incubated with expanded primary NK cells for the indicated times. Apoptosis of CD34^+^ CML-BC cells in the absence (dotted line) or presence (solid line) of NK cells was measured as in (**a**). The percent apoptosis of CD34^+^ CML-BC cells is shown as a line graph. (**c**) Representative flow cytometric profiles showing the surface expression of NKG2D ligand (MICA/B, ULBP1, ULBP2/5/6, and ULBP3), DNAM-1 ligand (PVR and Nectin-2), or NKp30 ligand (B7-H6) on gated CD34^+^ CML-BC cells (red histogram). The representative mean fluorescence intensity (MFI) of each ligand in the histogram is presented as the result of two independent experiments. (**d**) The MFI of the expression of each indicated ligand on gated CD34^+^ CML-BC cells (*n* = 5) is shown. Horizontal bars indicated the medians. (**e**,**f**) MNCs from a CML-BC patient were incubated with expanded primary NK cells preincubated with mAb to the indicated receptors (20 μg/mL) for 2 h. Apoptosis of CD34^+^ CML-BC cells was measured as in (**a**). Shown are representative flow cytometry profiles (**e**) and a graph of statistical bar charts (**f**), demonstrating the percentage of apoptotic CD34^+^ CML-BC cells. The mean values ± s.d. of two independent experiments are shown. * *p* < 0.05 by Student’s *t*-test.

**Figure 4 cancers-12-01923-f004:**
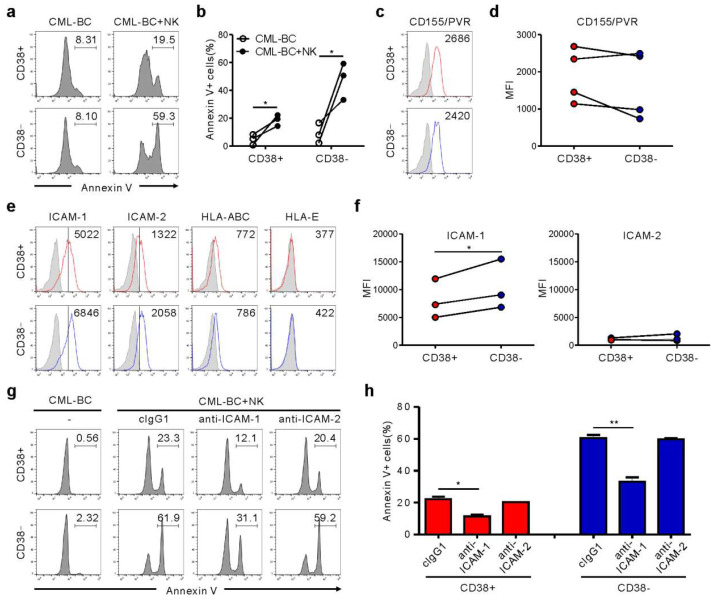
NK cells trigger preferential cytolysis of the CD34^+^CD38^−^ subset in CML-BC blasts through an interaction with ICAM-1. (**a**,**b**) Apoptosis of CD38^+^ vs. CD38^−^ subsets on gated CD34^+^ CML-BC cells (*n* = 3) in the absence or presence of NK cells, as measured by annexin V-FITC, anti-CD34-PE, and anti-CD38-APC staining. Shown are representative flow cytometry profiles (**a**) and a matched line graph (**b**) demonstrating the percentage of apoptotic CD34^+^CD38^−^ vs. CD34^+^CD38^−^ cells. (**c**,**d**) Similar surface levels of CD155/PVR between CD38^+^ and CD38^−^ subsets on gated CD34^+^ CML-BC cells (*n* = 4). Shown are representative flow cytometry profiles (**c**) and a matched line graph (**d**) demonstrating the MFI of PVR expression on CD34^+^CD38^−^ vs. CD34^+^CD38^−^ cells. (**e**) Representative flow cytometric profiles showing the surface expression of adhesion ligand (ICAM-1 and ICAM-2) or inhibitory ligand (HLA-ABC and HLA-E) on CD34^+^CD38^+^ vs. CD34^+^CD38^−^ subsets. The representative mean fluorescence intensity (MFI) of each ligand in the histogram is presented as the result of two independent experiments. (**f**) Upregulated levels of ICAM-1 on the CD34^+^CD38^−^ subset compared with CD34^+^CD38^+^ subsets (*n* = 3). Shown are matched line graphs demonstrating the MFI of ICAM-1 and ICAM-2 expression on CD34^+^CD38^−^ vs. CD34^+^CD38^−^ cells. (**g**,**h**) MNCs from a CML-BC patient were incubated with expanded primary NK cells preincubated with mAb to the indicated receptors (20 μg/mL) for 2 h. Apoptosis of CD38^+^ vs. CD38^−^ subsets on gated CD34^+^ CML-BC cells was measured as in (**a**). Shown are representative flow cytometry profiles (**g**) and a graph of statistical bar charts (**h**) demonstrating the percentage of apoptotic CD34^+^CD38^−^ vs. CD34^+^CD38^−^ cells. The mean values ± s.d. of two independent experiments are shown. * *p* < 0.05; ** *p* < 0.01 by Student’s *t*-test.

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
