# Peer review of "PVR and ICAM-1 on Blast Crisis CML Stem and Progenitor Cells with TKI Resistance Confer Susceptibility to NK Cells"

_cancers, 2020, doi:10.3390/cancers12071923_

Round 1
Reviewer 1 Report
This manuscript descripts the potential BCR-ABL1 mutation in clinical CML patients which caused TKI resistance and intolerance. The authors found human CML-BC cells harboring a T315I mutation reacted a less sensitivity to imatinib treatment and NK cell degranulation. And furthermore, several patients harbored single mutation (T315I, E255V, or E255K), one had compound mutation (F317L/E275K), and two had no 130 mutation in BCR-ABL1 gene were also tested and confirmed this finding. This study is well designed and the figures are well presented. From my point of view, this manuscript is fully qualified to be accepted in brief report in Cancers.Author Response
This manuscript descripts the potential BCR-ABL1 mutation in clinical CML patients which caused TKI resistance and intolerance. The authors found human CML-BC cells harboring a T315I mutation reacted a less sensitivity to imatinib treatment and NK cell degranulation. And furthermore, several patients harbored single mutation (T315I, E255V, or E255K), one had compound mutation (F317L/E275K), and two had no 130 mutation in BCR-ABL1 gene were also tested and confirmed this finding. This study is well designed and the figures are well presented. From my point of view, this manuscript is fully qualified to be accepted in brief report in Cancers.
Re) We thank the reviewer for appreciation of our work as being acceptable for publication in Cancers. In the revision, we incorporated the details of IRB approval for patient samples (ethical code, the date of approval) as the assessment of their susceptibility to NK cells is the central part of the paper. [Line 218]
Reviewer 2 Report
NK cells have been shown in vivo to correlate whit better prognosis in CML patients and it has been hypothized they could contribute to CML immune surveillance particularly in the case of minimal residual disease, that is often due to the persistance of TKI-resistant LSC. The present paper provides scientific demostration of possible mechanisms that could be involved in the ability of NK cells to eradicate the disease and, importantly, provide an interesting prespective of how novel immunotherapeuthics approaches can be design to overcome the CML-TKI resistance in CML patients. The paper is well organized and written.
Major point:
1)Despite authors claim that they isolate CML blasts from patient characterized either by mutation leading to TKI resistance either from patients still susceptible to TKI treatment ( i.e. with no relevant mutations), it appears that they analyzed NK cell cytolytic acitvity only against TKI resisistant CML blasts. However, It would be of interest to evaluate NK cell activity also against CML blasts still susceptible to TKI treatment because such comparison could be important to underline the potential crucial role of NK cells in TKI resistant patients
Minor point:
Figure S2. There must be a misunderstanding: data according to the text and Figure title should refer to K562 cell line and not to KCL22 and KCL22-IM
Author Response
1) Despite authors claim that they isolate CML blasts from patient characterized either by mutation leading to TKI resistance either from patients still susceptible to TKI treatment (i.e. with no relevant mutations), it appears that they analyzed NK cell cytolytic acitvity only against TKI resistant CML blasts. However, it would be of interest to evaluate NK cell activity also against CML blasts still susceptible to TKI treatment because such comparison could be important to underline the potential crucial role of NK cells in TKI resistant patients.
Re) In addition to point mutations in BCR-ABL1 kinase domain, no BCR-ABL1 mutation is observed in a significant proportion of TKI-resistant CML-BC patients, which involves the activation of alternative oncogenic pathways or leukemic stem cells as BCR-ABL1-independent mechanisms. In our study, NK cells consistently triggered cytolysis of TKI-resistant CML blasts with no BCR-ABL1 mutation (indicated as “Not detected” in Figure 3b) as well as diverse BCR-ABL1 mutations. Thus, we anticipate that TKI-sensitive CML blasts with no relevant BCR-ABL1 mutation would be equally susceptible to NK cells as were their surrogate KCL-22 cells (Figure 1b). Unfortunately, patients with TKI-sensitive CML blasts are routinely subjected to clinical TKI treatment and not accessible under our IRB approval that limits the study to the advanced, TKI-resistant CML with dismal outcome. Accordingly, the requirement for further study on the comparison of susceptibility to NK cells between TKI-sensitive and –resistant CML blasts, together with more explanation on the relation between TKI-resistance and BCR-ABL1 mutation, have been discussed in the revised manuscript. [Line 51; Lines 139~141]
Minor point:
Figure S2. There must be a misunderstanding: data according to the text and Figure title should refer to K562 cell line and not to KCL22 and KCL22-IM
Re) We thank the reviewer for commenting this mistake, which has been corrected in the revised legend for Figure S2 (change of “KCL-22 or KCL-22M” to “K562”).
Round 2
Reviewer 2 Report
The authors aswered appropriately to reviewer's concerns, thus it appears that the paper is now suitable fo publication on Cancers.